# Lipid Oxidation Products and the Risk of Cardiovascular Diseases: Role of Lipoprotein Transport

**DOI:** 10.3390/antiox13050512

**Published:** 2024-04-24

**Authors:** Markku Ahotupa

**Affiliations:** 1Centre for Population Health Research, University of Turku and Turku University Hospital, 20520 Turku, Finland; ahotupa@utu.fi; 2Research Centre of Applied and Preventive Cardiovascular Medicine, University of Turku, 20520 Turku, Finland

**Keywords:** atherosclerosis, cardiovascular diseases, cholesterol, high-density lipoprotein, lipid oxidation, lipoprotein functions, low-density lipoprotein, risk factors

## Abstract

Cholesterol has for decades ruled the history of atherosclerotic cardiovascular diseases (CVDs), and the present view of the etiology of the disease is based on the transport of cholesterol by plasma lipoproteins. The new knowledge of the lipoprotein-specific transport of lipid oxidation products (LOPs) has introduced another direction to the research of CVD, revealing strong associations between lipoprotein transport functions, atherogenic LOP, and CVD. The aim of this review is to present the evidence of the lipoprotein-specific transport of LOP and to evaluate the potential consequences of the proposed role of the LOP transport as a risk factor. The associations of cholesterol and lipoprotein LOP with the known risk factors of CVD are mostly parallel, and because of the common transport and cellular intake mechanisms it is difficult to ascertain the independent effects of either cholesterol or LOP. While cholesterol is known to have important physiological functions, LOPs are merely regarded as metabolic residues and able to initiate and boost atherogenic processes. It is therefore likely that with the increased knowledge of the lipoprotein-specific transport of LOP, the role of cholesterol as a risk factor of CVD will be challenged.

## 1. Introduction

Plasma lipoproteins play a significant role in the steering of lipid metabolism, both as the conveyors of lipophilic substances in blood, and as the mediators of the receptor-mediated cellular uptake of lipids [1]. It is well known today that the two main lipoprotein classes, low-density lipoprotein (LDL) and high-density lipoprotein (HDL), transport plasma lipids to opposite directions. LDL is the main carrier of cholesterol to peripheral tissues, while the reverse transport by HDL returns excess tissue cholesterol to the liver. Likewise, it is also known that LDL and HDL have independent and opposing roles as risk factors for cardiovascular diseases (CVDs). While LDL is associated with an elevated risk of CVD, high HDL concentration appears to be protective. The different roles of LDL and HDL as risk factors are explained by the lipoprotein specificity of the transport and receptor-binding functions, which have been worked out in studies over the past 60 years [1]. 

The formation of atheromatous plaques in arterial walls is a key element in the development of atherosclerosis. The plaques have a complex histology, the core of which consists of lipids, chiefly cholesteryl esters, accumulated in the arterial intima. A high level of plasma cholesterol, on the other hand, is undoubtedly a significant risk factor of atherosclerosis. These facts, recognized early in the history of atherosclerosis research, have been for decades the cornerstones for the undisputed dominant role of cholesterol in the science and management of CVD. 

After the “discovery” of the oxidized LDL, it soon became a target for the investigations of the molecular mechanisms of atherosclerosis, and the various ways by which the oxidized LDL may contribute to atherogenesis became evident. Based on this development, the idea of “the oxidation theory of atherosclerosis” was presented in 1989 [2]. 

Lipid oxidation products (LOPs) are a heterogenous group of chemical compounds arising in the non-physiological oxidation of lipids, and generally seen as metabolic residues with potentially adverse biological effects. The presence of LOP in plasma lipoproteins was first detected several decades ago [3,4,5,6,7]. The physiological significance of the lipoprotein-specific LOP transport was recognized much later, as it turned out that plasma lipoproteins are the active carriers of LOP. Similarly to the transport of cholesterol, LDL and triglyceride-rich lipoproteins (TRLs) direct the movement of LOP towards peripheral tissues, while the role of HDL appears to be the removal of LOP in a reverse transport [8,9,10].

LOPs impair cellular functions in many ways and are well known from their multiple pro-atherogenic effects [11]. Among other oxidants, reactive LOP can bring forth oxidative modifications in the structure of LDL, thereby causing a deviation from the normal metabolism and initiating the formation of foam cells and atherosclerotic plaques. LOPs contribute not only to the formation oxidative modifications, but also amplify by multiple mechanisms the subsequent stages of atherosclerosis progression (described in more detail in next chapter; Figure 1).

The obvious atherogenic potential of LOP, together with the emerging new information on the lipoprotein-specific LOP transport, raise the question of their potential impact on the risk of atherosclerosis and CVD. The aim of this review is to present data from studies on lipoprotein LOP and CVD, discuss how this comports with the present conception of the etiology of the disease, and speculate the possible consequences regarding the prevention, diagnosis, and management of CVD.

## 2. Oxidized LDL and Atherosclerosis

The oxidation theory of atherosclerosis is based on the fact that oxidative modification in the structure of the LDL leads to an inability to bind to the LDL receptor (LDLR), and thereby hampers the normal metabolism of the lipoprotein. Scavenger receptors (like scavenger receptor A-1 and other scavenger receptors that were found later), on the other hand, bind avidly oxidized LDL, and this deviation of the LDL metabolism leads to the progressive transformation of scavenger receptor-bearing cells (macrophages and smooth muscle cells) into foam cells [12].

The mechanism whereby the native LDL becomes a substrate for the scavenger receptor pathway is explained by the initial oxidation of LDL lipids, and a subsequent modification of apolipoprotein B by reactive LOP [11]. The reaction of LOP with the lysine groups in apolipoprotein B is postulated to be the reason for the altered receptor binding, since lysine residues are known to be required for LDL recognition by the LDLR [12].

Since the presentation of the idea of the oxidation theory of atherosclerosis, it has been generally assumed that the oxidation of the LDL takes place within the vessel wall and is caused by the endogenous free radical reactions induced, e.g., by metal ions, superoxide anion, nitric oxide, lipoxygenase, or myeloperoxidase [11].

LDL oxidation contributes to atherogenesis not only by steering the metabolism towards scavenger receptor-bearing cells. Indeed, oxidized LDL has a wide spectrum of biological effects that may substantially increase its total pro-atherogenic impact [11,13]. Oxidized LDL is known to be cytotoxic and can cause endothelial injury or dysfunction allowing an increased LDL deposition in the subendothelial space. Moreover, oxidized LDL is known to stimulate the production of the monocyte chemotactic protein-1 in vascular cells, which leads to the recruitment of inflammatory cells; this is generally regarded as one key point in the pathogenesis of atherosclerosis. The injurious effects of the recruited inflammatory cells may even be amplified by oxidized LDL, since it is known to be chemostatic for tissue macrophages and can induce the expression of the scavenger receptors of macrophages. Yet another way through which oxidized LDL may be involved in atherogenesis is the mitogenic effect for smooth muscle cells. By this mechanism, oxidized LDL can directly influence the formation of plaques and thickening of the vessel wall.

It is important to note that oxidized lipids in LDL appear to be responsible for most, if not all, of the atherogenic effects, since LOPs have been shown to cause the toxic, inflammatory, and mitogenic effects described for oxidized LDL (Table 1). Oxidatively modified apolipoprotein B, in turn, is merely known as the cause for the disturbed LDLR-mediated cellular uptake of LDL.

## 3. The Protective Role of HDL

With advancement of the study of HDL it has come out that besides lipid transport, HDL appears to have several other physiological functions (Table 2). These include numerous examples of communication with cells as a part of vital cellular functions, and the inactivation and removal of biohazards like bacterial toxins (for a recent review, see [14]).

Rather than on cholesterol carried by HDL, the various HDL functions are based on the activities of the entire HDL particles or distinct proteins or lipids on HDL [14]. The diversity of the physiological functions of HDL has been explained to result from the heterogeneity of the composition of the HDL particles. The prominent heterogeneity of HDL, which has come into daylight within the last two decades, has been somewhat surprising even to scientists working with this subject: the number of different HDL-associated proteins is estimated to be over 300, and the number of associated lipids may be even higher [14,15].

The concentrations of the distinct proteins and lipids on HDL may differ markedly from each other ranging from sub-micromolar (“low-abundant”) to millimolar (“high-abundant”). The low-abundant HDL proteins, in turn, can be categorized either as lipoprotein-specific (e.g., lecithin:cholesterol acyltransferase, cholesteryl ester transfer protein, and paraoxonase-1) or ancillary proteins. Despite their low abundance, ancillary proteins and low-abundant lipids are also known to have important biological functions [14,16].

The concentration of plasma HDL-cholesterol has for decades been regarded as an indicator of the risk of atherosclerosis and CVD. In addition to CVD, a low level of HDL-cholesterol is known to be related to diabetes, chronic kidney disease, sepsis, and the risk of infectious diseases. More recent findings indicate, however, that for the estimation of the risk of CVD the HDL-cholesterol value may not be the best measure, and new evidence has questioned a causal association between HDL-cholesterol and CVD [14,16]. In line with this, the cholesterol efflux capacity, which is indicative of the HDL functionality, has strong inverse association with the indicators of atherosclerosis, and these associations are independent of the total level of HDL-cholesterol [17]. These findings underline the significance of the HDL-mediated macrophage cholesterol efflux and have prompted a need for clinically usable biomarkers for the estimation of the efficacy of the efflux capacity. In addition to the cholesterol efflux capacity, there are also other functional measures of HDL, such as the number of HDL particles or distinct HDL proteins, which has proved to be a better predictor of CVD than the HDL-cholesterol [14]. It is obvious, as concluded by von Eckardstein et al. [14], that in clinical studies of the role of HDL only the biomarkers directly related to essential pathological mechanisms should be used.

## 4. Transport of Lipophilic Substances Other Than Plasma Lipids

In addition to the plasma lipids and LOP, lipoproteins act as carriers for a wide variety of other lipophilic compounds in blood. For example, the fat-soluble vitamins and non-nutrient lipophilic substances (phytochemicals, drugs, and xenobiotics) that enter the alimentary tract can be absorbed in the intestine and incorporated into lipoproteins along with the dietary fat. These compounds, too, can be taken up by cells via the receptor-mediated uptake mechanisms [18,19,20,21,22,23,24]. The physiological significance of the lipoprotein-dependent transport and the uptake of substances other than the dietary lipids has received little attention. It has been reported, however, that oxidized lipids in LDL are incorporated into the cells and detected in cell extracts [25]. Moreover, biologically active phytochemicals, incorporated into LDL and internalized by cultured cells via the LDLR pathway, have been shown to retain their biological activity [21].

Experimental studies with LDLR-overexpressing and -knockdown cells showed that LDLR regulates the behavior of drugs associated with VLDL/LDL [24]. The same report further showed that the association with lipoproteins can affect the transport, metabolism, and efficacy of lipophilic drugs in humans. Consequently, the lipoprotein-mediated drug transport is a potential contributor to the transport and metabolism of lipophilic drugs in the body, the role of which should be considered when developing new pharmacotherapies [24]. In line with this, the evidence of the lipoprotein transport of fat-soluble xenobiotics would mean that lipoproteins are involved in the distribution and toxic effects of these chemicals in the body [26].

## 5. Lipid Oxidation Products in Human Plasma

LOPs arise during the progression of the peroxidation of polyunsaturated fatty acids (Figure 2). Peroxidation reactions are induced by reactive free radicals and oxidant stress and can generate a wide variety of LOPs of different structures. In the first phase of the stepwise chain of reactions, intermediary derivatives (e.g., hydroxides, hydroperoxides, and epoxides) are formed. With advancement of the oxidative breakdown, these intermediary derivatives can give rise to the formation of reactive aldehydes and less reactive products like ketones and alkanes (Figure 2).

Reactive aldehydes, in turn, can react and form adducts with cellular macromolecules, notably with proteins, giving rise to advanced lipid peroxidation end products that are known to cause protein modifications and disturb cellular functions [27,28]. The peroxidation of PUFA can also lead to the formation of oxidatively modified lipids with biological activity, such as the oxidized phospholipids and isoprostanes [27].

LOP in human plasma can have both endogenous and exogenous origins. The endogenously formed LOP result from oxidative stress in the body, which may be due, e.g., to physical exercise, inflammation, irradiation, or exposure to chemicals [29,30]. Physical exercise has even proved to be a useful physiological model to investigate the formation and distribution of endogenous LOP [9]. Exogenous LOP, in turn, enters the body through the alimentary tract along with the ingested food. It has been shown that LOPs in food are absorbed in the intestine and incorporated into plasma lipoproteins together with the dietary lipids [5,6,7,9]. The average daily intake of LOP associated with dietary fat has been estimated to be more than 1 mmol per day [31].

LOP in food may arise from the enzymatic oxidation, photo-oxidation, autoxidation, or thermal oxidation of food lipids, which can take place during the storage and processing of food. In addition, it has come out that the peroxidation of food lipids occurs also in the alimentary tract during digestion [32]. Food LOP comes from various lipid classes (e.g., triglycerides, phospholipids, and sterol esters), and can be present attached to the original lipid (glycerol or sterol) backbone, or as free molecules. Food-contained LOPs are shown to be absorbed and incorporated into plasma lipoproteins, including LDL and HDL [5,6,7,9]. In all, this indicates that food LOP appears as another type of dietary risk factors, emphasizing the importance of the pro/antioxidant effects of nutrients.

The potential significance of food as a source of LOP was clearly shown in a study, where young healthy male volunteers consumed pan-fried and sous-vide thermally processed beef rump steaks. The experimental meals were prepared replicating real cooking conditions, which gave tender or medium rare steaks. Twenty-three oxidation-modified lipid structures were detected from the beef, and fifteen of these were also found in the postprandial plasma [33].

## 6. Lipoprotein-Specific Transport of LOP

The transport of LOP by plasma lipoproteins has received little attention, and the presence of LOP in circulating lipoproteins has mainly been regarded to result from endogenous oxidative stress. Accumulating evidence suggests, however, that the presence of LOP in plasma lipoproteins is not a random consequence of oxidative stress, but rather indicative of specific LOP transport functions. It appears that plasma lipoproteins are the active carriers of LOP, with LDL directing the transport toward peripheral tissues and HDL being active in the reverse transport. The biological mechanisms responsible for the lipoprotein-specific transport and cellular uptake of the native (unoxidized) lipids may at the same time lead to lipoprotein-specificity in the transport and cellular uptake of LOP.

The evidence of the lipoprotein-specific LOP transport comes mainly from the studies of lipoprotein LOP transport during physiological oxidative stress and is supported by the studies of the functional apolipoprotein A-1 (Apo A-1) mimetic peptides.

### 6.1. Studies on Physiological Oxidative Stress in Human Volunteers

Acute increases in plasma LOP concentrations are known to be caused, e.g., by LOP-containing meals [5,6,7,9] or by strenuous physical activity [30]. Both these physiological circumstances have been used as experimental models to investigate the lipoprotein LOP transport (Figure 3) [9]. Food LOP, typically found in fatty meals, represents an exogenous source of LOP for the human body, while the LOP generated as a result of physical exercise-induced oxidative stress is an example of endogenously formed LOP.

The roles of lipoproteins as the carriers of exogenous LOP in human body were investigated in a model with a common hamburger meal (available in fast-food restaurants worldwide) as the “high-fat meal” [9]. Altogether, over 100 volunteers participated in these studies, in which the concentration of LOP in plasma lipoproteins was monitored during the postprandial period following the hamburger meal [9,34].

The results consistently showed that the concentration of LOP in triglyceride-rich lipoproteins (TRLs and LDL) started to rise within 1 h after the meal. The LOP levels in TRL and LDL reached maximal levels within 2–4 h, after which they gradually returned to pre-meal values. It is worth noting that even though the concentration of LOP in LDL did increase, the concentration of LDL-cholesterol did not change. Moreover, detailed studies showed that the fatty meal-induced elevation in lipoprotein LOP was not due to increased oxidative stress, nor to altered antioxidant functions [9]. Unlike the case of TRL and LDL, there was only a small and delayed increase in HDL-LOP concentration 6 h after the meal [9]. The uneven (lipoprotein-specific) postprandial distribution of LOP between the lipoprotein classes showed that food-derived LOP are not randomly incorporated into various lipoproteins.

In contrast to the substantial rise in LDL-LOP, the level of oxidatively modified apolipoprotein B has been reported to decrease after a high-fat meal [35]. This finding supports the view that the fatty meal-induced increase in LDL-LOP is not due to the oxidation of lipids within the LDL particle.

Among several other indicators of oxidative stress, strenuous physical exercise is well known to increase the level of LOP in human plasma [30], and the follow-up of the concentrations of LOP in lipoproteins during and after the exercise has allowed the investigation of the transport of endogenously formed LOP [9].

Opposite to the effect of a fatty meal, strenuous physical exercise has little, if any, effect on the concentration of LOP in LDL (Figure 3). Contrary to this, physical exercise results in a substantial increase in the concentration of LOP in HDL. This effect may last for hours after a single bout of aerobic exercise, and it appears to depend on the intensity of the physical activity [9]. In a study with well-trained athletes (middle-distance and marathon runners), it was found that the exercise-induced increase in LOP in HDL is related to the training history of the athletes [36].

The increase in LOP in HDL particles is not due to the oxidation of the lipids carried by HDL, since no such increase is seen in LDL despite the lower oxidation resistance of LDL [37]. In line with this, Shao and Heinecke [8] have concluded that the oxidation of either HDL or LDL in plasma is highly unlikely. Moreover, it seems unlikely that LOP could be transferred directly from LDL to HDL in plasma, since the transfer of LOP between lipoproteins appears to be too slow to affect their distribution in lipoproteins [3].

Taken together, the studies on physiological oxidative stress point to the lipoprotein-specificity of the transport of LOP in circulation: LOP in food are absorbed and incorporated into serum TRL and LDL, directing the flow of LOP towards peripheral tissues; HDL appears to have an opposite transport function, and can respond to oxidative stress by substantially increasing the reverse transport of LOP.

### 6.2. Experimental Evidence of the Role of HDL

More evidence supporting the idea of LOP scavenging, transport, and clearance function of HDL comes from studies with apo A-1 mimetic peptides and apolipoprotein M (apo M). Similarly to the atheroprotective functions of HDL, apo A-1 mimetic peptides are known to have positive effects on lipid metabolism, oxidative stress, inflammation, and insulin resistance [38,39,40]. Studies by Van Lenten et al. [38] showed that apo A-1 mimetics have a remarkable binding affinity for LOP. It is of special interest that the LOP scavenging and removing capacity of apo A-1 mimetics was found to be strongly related to positive health effects [38].

Studies with the “4F peptide” as a representative of the apo A-1 mimetics has given additional evidence regarding the postulated link between the clearance of LOP and atheroprotection. While decreasing the hepatic and plasma concentrations of LOP, the “4F peptide” also causes a significant reduction in atherosclerotic lesions [39,40]. In line with this, the ability of the HDL-associated apo M to trap oxidized phospholipids appears to be associated with its atheroprotective effects [41].

Studies on a perfused rat liver model have shed light on the role of HDL in the clearance of LOP from the body. Early studies by Stocker and coworkers [42] showed that the liver removes LOP from HDL, but not from LDL particles. In support of the role of HDL in the clearance of LOP, Fluiter et al. [43] further reported that there is a preferential liver uptake, coupled to a rapid biliary excretion pathway, of oxidized cholesteryl esters in HDL as compared with the unoxidized cholesteryl esters.

A recent study showed that HDLs can utilize their capacity of loading themselves with lipophilic compounds, similarly to their ability to extract cellular cholesterol, to reduce the cell content of lipophilic drugs and xenobiotics. ABC transporters were not required for the process, but however, could favor it [26].

Together with the evidence of the intervention studies in human volunteers, results of the experimental studies suggest that the transport of LOP by HDL is part of a protective transport function, where HDL protects peripheral vessels and tissues from LOP by transporting these potentially toxic substances to the liver, as originally proposed for cholesterol [8,9,10].

## 7. Potential Significance of Lipoprotein LOP Transport for Oxidative Stress

Studies of the LOP transport during physiological oxidative stress show that LDL carries LOP to peripheral tissues, and HDL is active in the reverse transport. Knowing the pro-oxidant potential of LOP, this would advocate a pro-oxidant role for the LDL transport, whereas the transport by HDL would be a part of the antioxidant defense of the body.

At present there is little direct evidence of a causal connection between LDL and oxidative stress, but indirect evidence has suggested a link between these two. The level of oxidative stress is reported to be elevated among patients with FH [44,45], and the suppressing of the oxidative stress has been speculated as a potential therapeutic means to reduce the atherogenesis in patients with FH [45]. In line with this, statin treatment has been reported to be associated with decreased oxidative stress [46,47], prompting the speculations of the contribution of the antioxidant properties of statins in their beneficial effects [47].

Besides the reverse cholesterol transport, HDL is known to have several other functions potentially affecting atherogenesis, and antioxidant activity has frequently been mentioned as one of these. The inactivation of lipid hydroperoxides and the hydrolysis of oxidized phospholipids are known examples of the LDL-specific antioxidant action of HDL. Apo A-1 and enzymes associated with the HDL particles are important components of HDL contributing to these mechanisms [10,48].

Like in the case of LDL, there is little direct evidence of a causal connection between HDL and oxidative stress in human body. Yet, as part of the studies on physiological oxidative stress and LOP transport, a positive association was found to exist between plasma HDL level and the clearance of the fatty meal-induced LOP load, which was suggested to be indicative of an antioxidant function of HDL in vivo [9,34].

## 8. Measurement of Lipoprotein LOP Transport

The common assays for “oxidized LDL” are based on (i) the measurement of oxidatively modified apolipoprotein B, (ii) autoantibodies to oxidized LDL, or (iii) LOP in LDL. It is obvious from focusing either on the protein part of the LDL particle, or on autoantibodies developed against the oxidatively modified LDL, that they are not indicative of the LOP transporting function of LDL. Instead, LOP found in the lipid part of LDL may be used for the estimation of the LOP transport by LDL. The present analytical techniques for the estimation of LOP in LDL are based on the determination of the oxidatively modified lipids or end products of lipid oxidation in LDL isolated from plasma. Common examples of these are conjugated dienes, fatty acid hydroperoxides, oxidized phospholipids, and malondialdehyde. The interest for LOP in HDL appeared much later than in the case of LDL. Therefore, there are far less studies conducted and data available regarding LOP in HDL.

Recent studies of the lipoprotein-specific transport of LOP are by large based on the measurement of conjugated dienes in the lipid fraction of LDL. LOPs, as measured by the diene conjugation assay, are distributed to all main lipid classes of LDL. This assay has a strong correlation with the assays for oxidized apolipoprotein B and autoantibodies against oxidized LDL [49,50].

Based on the direction-specificity of the lipoprotein transport, it has been proposed that the level of LOP in LDL (LDL-LOP) would be indicative of a potential exposure of the peripheral vessels and tissues to LOP, and the concentration of LOP in HDL (HDL-LOP) would indicate a reverse transport of LOP toward the liver. Accordingly, the ratio of LDL-LOP/HDL-LOP would show the direction of the net movement of LOP in the body [9].

## 9. Lipoprotein LOP and the Risk of CVD

Cholesterol is known to be involved in many physiological functions, while the LOP appears to be metabolic residues with no physiological role. Cholesterol per se is not regarded as the progenitor of the chain of events leading to the development of atherosclerosis. On the other hand, oxidized lipids in LDL are involved in all major pathophysiological processes leading to the thickening and stagnation of arteries (Table 1). Therefore, the fact that LOP are transported and taken up by cells together with cholesterol has raised the question of a possible role of the lipoprotein LOP transport in the development of atherosclerosis and CVD. As an attempt to answer this question, the connections of lipoprotein LOP with known atherosclerosis risk factors, metabolic disorders related to atherosclerosis, subclinical atherosclerosis, and clinically verified atherosclerosis have been investigated in studies with human volunteers (Table 3).

### 9.1. Common Risk Factors

Age and gender are well known factors related to the risk of atherosclerosis and CVD. The incidence of diseases associated with atherosclerosis typically increases with age and is higher in men than in women. There are only a few studies where the dependence of lipoprotein LOP on either age or gender have been investigated. A substudy of The Cardiovascular Risk in Young Finns Study, with a total of 1395 subjects (629 males and 766 females), is by far the most comprehensive of these [50]. In addition to the age and gender, the associations of lipoprotein LOP with several other atherosclerosis risk factors were investigated in this multicenter follow-up study.

A significant sex difference was found to exist in both LDL-LOP and HDL-LOP concentrations: compared to men, the values of LDL-LOP were lower and HDL-LOP higher in female subjects [50]. Despite the relatively narrow age range of participants (24–39 years old at the time of blood sampling), age was associated in multivariate models with both LDL-LOP (direct association) and HDL-LOP (inverse association) [50]. It is worth mention that the age dependence of HDL-LOP appears to be opposite to that for the HDL-cholesterol, which has been reported to increase with age [51].

Of the common lipid risk factors of atherosclerosis, total cholesterol, LDL-cholesterol, and triglycerides showed strong correlation with LDL-LOP, but not with HDL-LOP [50]. The positive correlations of LDL-LOP with plasma lipids have been reported also in several previously published studies [49,52].

Blood pressure is another factor strongly related to the risk of atherosclerosis. As an indication of a possible association between lipoprotein LOP and blood pressure, an increased level of LDL-LOP has been reported in young men with borderline hypertension. In this study, ambulatory recording of the blood pressure over a 24 h period was employed [53]. In line with this, a bivariate association between the diastolic blood pressure and LDL-LOP was found in The Cardiovascular Risk in Young Finns Study [50].

The effect of tobacco smoking on lipoprotein LOP has been investigated in two studies. The participants of the first study (n = 164) were obtained from a population cohort of Finnish men aged 40–70 years. In this study, smokers had significantly higher levels of LDL-LOP (21%) than non-smokers [54]. In The Cardiovascular Risk in Young Finns Study daily smoking was found to be inversely associated with the HDL-LOP level in women [50].

### 9.2. Physical Activity

Physically active lifestyle is well known have beneficial effects on cardiovascular health reducing the overall risk of coronary heart disease and stroke [55,56]. During the past three decades, several studies have been carried out to investigate the effects of physical activity on lipoprotein LOP. The investigations range from cross-sectional studies of the effects of physically active lifestyle to intervention studies on the various forms of acute or long-term physical activity. Since the assay for HDL-LOP was taken 15 years later than that for the LDL-LOP, the data concerning HDL-LOP is missing from the studies published prior to 2010.

The first study reporting the effects of a physically active lifestyle was performed with veteran endurance athletes participating competitively in orienteering, running, cycling, or triathlon. The athletes were compared with matched subjects of the same age and socioeconomic status. It was found that the athletes had significantly lower LDL-LOP values than control subjects, and that LDL-LOP correlated inversely with the leisure intensive activity score (MET_int_; an indication of intensive physical exercise energy expenditure) [49].

More evidence of the decrease in LDL-LOP by physical activity was obtained from an intervention study where sedentary middle-aged men and women (n = 104), recruited by the local occupational health service, participated in a 10-month exercise program. The exercise intervention produced favorable changes in lipid profiles, and reduced the LDL-LOP by 23% and 26% in men and women, respectively. The ratio of LDL-LOP/LDL-cholesterol was also decreased, and it was 14% in men and 18% in women [49]. Another long-term follow-up study was performed with marathon runners (n = 127) who were preparing for a marathon race [57]. Venous blood samples were taken at four different time points during the study: 3 months (training period) and 6 days (final preparation period) prior to the marathon run, and on the marathon day 2–4 h before and immediately after the race. During the 6 day preparation period, subjects decreased training and increased the carbohydrate intake. Interestingly, the training period did not affect LDL-LOP values, but after the preparation period the values were substantially increased. Then, again, the marathon run restored the LDL-LOP values towards the level before the preparation period [57].

The effect of prolonged, low intensity exercise on LDL-LOP was investigated in an experimental setting, where healthy well-trained men performed a walk exercise on two consecutive days (6 h per day) [58]. Like in the above studies, the prolonged physical activity resulted in a decrease (−25%) in the level of plasma LDL-LOP. Favorable changes were seen also in the conventional plasma lipids, while the serum LDL-cholesterol (−14%) and triglycerides (−22%) decreased, and the HDL-cholesterol (+9%) increased [58].

Only one published study has reported the effects of a physically active lifestyle on HDL-LOP. In this study, a program of 6-month supervised physical activity increased the concentration of HDL-LOP and the ratio of HDL-LOP/HDL-cholesterol among elderly women. The concentration of LDL-LOP remained unchanged [59].

The effects of acute strenuous physical activity have been investigated in long-distance runs and treadmill exercises. During a long-distance (31 km) run, the total amount of LOP in whole serum increased but the concentration of LDL-LOP remained unchanged [60,61].

The finding that the long-distance run increased the amount of LOP in whole serum but not in LDL raised a question of the carrier of LOP generated during physical activity. This topic was later studied in more detail, and like in the previous studies acute physical exercise did not affect the LDL-LOP level. Contrary to this, HDL-LOPs were substantially increased, and remained elevated for hours after the exercise. In addition to the total amount of HDL-LOP, the ratio of HDL-LOP/HDL-cholesterol was also increased [9]. This was a significant novel discovery pointing at the LOP-clearing function of HDL [56,62,63].

### 9.3. Obesity and Weight Reduction

Overweight and obesity are associated with significantly increased morbidity and mortality from the CVD [56,62,63]. Consequently, weight reduction is among the first lines of defense for patients with an increased risk of developing atherosclerosis and the resulting CVD.

The relationships of LOP with body weight and waist circumference were investigated in The Cardiovascular Risk in Young Finns Study. It was found that the concentration of LDL-LOP has significant bivariate associations with both BMI and waist circumference [50]. In accordance with this, Sorokin et al. [64] reported associations between the lipoprotein LOP and BMI (positive association with LDL-LOP and negative association with HDL-LOP).

The effect of weight reduction has been investigated in three independent intervention studies. In the first study, the effects of a 12-week weight-reduction and a subsequent 9-month weight-maintenance program on LDL-LOP were investigated in a group (n = 77) of obese premenopausal females [65]. A 13 kg decrease in body weight was associated with a 40% decrease in the concentration of LDL-LOP. The weight reduction correlated with the decrease in LDL-LOP. The concentration of LDL-cholesterol was also decreased, but to a smaller extent (11%). Altogether, this was the first study to investigate the effects of obesity and weight reduction on oxidized LDL.

In another study, performed with obese men (n = 68), a 2-month weight-reduction period was followed by a 6-month weight-maintenance program, after which participants of the study were followed up for another 2 years period [66]. The mean weight loss at the end of the weight-reduction period was 14.5 kg (14%), which was associated with a 22% decrease in LDL-LOP concentration. At the end of the 2-year follow-up, the weight regain from the end of the weight-reduction period in whole study group was 11% and regain in the concentration of LDL-LOP was 30%. To study the effect of successful weight maintenance, the study population was divided into two subpopulations regarding the success in maintaining the weight loss. It was found that the regain of weight was firmly associated with an increase in LDL-LOP concentration [66].

Cardiorespiratory fitness has been shown to protect obese individuals from cardiorespiratory mortality [67]. In accordance with this, overweight men with good aerobic or muscular fitness were found to have lower levels of LDL-LOP than unfit men of the same weight [68]. This finding further strengthened the view of a connection between LDL-LOP and the risk of atherosclerotic diseases.

### 9.4. Dietary Factors

Despite the fact that nutritional aspects are well-known contributors to the risk of atherosclerosis and CVD, the number of studies focused on nutrition and lipoprotein LOP has remained low.

In a cross-sectional study setting, it was found that serum polyunsaturated fatty acids (PUFAs), and in particular the n6 PUFA proportion, were negatively associated with LDL-LOP, whereas the associations of saturated (SFAs) and monounsaturated fatty acids (MUFAs) were positive [69]. These findings at the population level are in line with the observations linking PUFAs with a lower and SFAs with an increased risk of CDV [70].

The finding of the positive effect of PUFA on LDL-LOP level is supported by results from an intervention study with modified Mediterranean-type diet rich in n3 PUFA. A 12-week intervention resulted in an 8% decrease in serum LDL-LOP level, and the concentrations of total and LDL-cholesterol decreased by 8% and 11%, respectively [71].

The effects of separate components of nutrition on lipoprotein LOP were investigated in two early intervention studies. The first of these focused on the effects of lycopene, a carotenoid available in tomatoes. In this study, the dietary supplementations doubled serum lycopene levels and significantly decreased the concentration of LDL-LOP, but had no effect on total, LDL-, or HDL-cholesterol concentrations [72]. In another study, hyperlipidemic subjects consumed whole almonds as snacks for 1 month. The consumption of almonds resulted in significant reductions in LDL-LOP (14%) and LDL-cholesterol (9%) concentrations [73].

Marniemi et al. [74] studied the effects of 8 weeks of antioxidant vitamin supplementation (100 mg α-tocopherol + 500 mg ascorbic acid per day) and found that even though the antioxidant supplementation significantly increased the antioxidant capacity of LDL particles, the concentration of LDL-LOP did not change (Figure 4). The same result was also obtained in another antioxidant intervention study with different antioxidant dosing and treatment period. Daily doses of α-tocopherol (294 mg), ascorbic acid (1000 mg), and ubiquinone (60 mg) for 4 weeks raised substantially the antioxidant capacity of LDL and the levels of antioxidant vitamins in blood but did not affect the level of LOP either in serum or LDL [61] (Figure 4). The fact that antioxidant supplementation does not affect serum LDL-LOP levels despite the substantial increase in LDL antioxidant capacity indicates that LOP carried by circulating LDL does not come from the peroxidation of the lipids within the LDL particles.

Similarly to the effects of antioxidant vitamins, flavonoid supplements in food failed to influence the LDL-LOP levels. Healthy male volunteers consumed flavonols (isorhamnetin, quercetin and kaempferol) daily for 4 weeks at a dose level that increased the concentrations of flavonols in plasma. Yet, serum LDL-LOP remained at the control level [75].

The studies described above have provided information regarding long-term dietary effects. In addition to these, the acute postprandial effects of dietary fat on lipoprotein LOP have been investigated (described in Studies on physiological oxidative stress in human volunteers). In short, these studies substantiate that LOP in food are absorbed and incorporated into plasma lipoproteins and transported in plasma similarly to the native (unoxidized) lipids. This underlines the potential importance of the lipoprotein transport of the food-derived molecular species with respect to the risk of CVD.

### 9.5. Metabolic Disorders Related to the Risk of CVD

Three clinical studies have focused on the co-existence of LDL-LOP and metabolic disorders related to the risk of CVD. In these studies, lipoprotein LOP was investigated in relation to insulin sensitivity, metabolic syndrome, and the development of fatty liver.

A 32-month lifestyle intervention study investigated whether changes in LDL–LOP concentrations are related to insulin sensitivity [76]. The intervention consisted of a weight-reduction program (2 months) followed by dietary and physical exercise counseling (6 months), and a follow-up period (2 years). The experimental protocol suited well for the purpose, since it caused substantial changes in insulin sensitivity (determined by the homeostasis model assessment of insulin resistance, HOMA-IR) of the participants. The detected changes in HOMA-IR were clearly parallel to those in LDL-LOP and LDL-LOP/HDL-cholesterol ratio, suggesting a connection between insulin metabolism and LDL-LOP [76].

Metabolic syndrome and lipoprotein LOP were investigated in a nationally representative sample of Finnish young men with and without the metabolic syndrome (MetS) [77]. The participants were divided according to the IDF 2007 criteria into MetS (n = 54) and non-MetS (n = 790), and age, smoking, and the leisure-time physical activity were used as covariates. The results showed that a high LDL-LOP level predicted MetS, and it was concluded that an elevated concentration of the LDL-LOP is a significant predisposing factor in the development of MetS [77].

Fatty liver is another metabolic disorder that is a risk factor for the CVD, and it is apparent that there are common progenitors for these pathological conditions [78]. Kaikkonen et al. [79] have studied the associations of the lipoprotein LOP with fatty liver assessed by ultrasonography. The participants of the study were 1286 middle-aged subjects with normal liver, and 288 subjects with fatty liver. After the adjustment for age, sex, leisure-time physical activity, body mass index, alcohol intake, smoking, serum LDL- and HDL-cholesterol as well as particle concentrations, the participants with elevated LDL-LOP were found to have an increased risk of a fatty liver. In particular, the high ratio of LDL-LOP/HDL-LOP indicated an increased risk. This relation was independent of the LDL and HDL particle concentrations and cholesterol content, other common risk factors for fatty liver, and inflammation. The LDL/HDL-cholesterol ratio was not associated with future fatty liver [79]. Based on these data, lipoprotein LOP appeared to be implicated in yet another pathophysiological process, the development of a fatty liver in middle-aged adults.

### 9.6. Subclinical Atherosclerosis

The clinical complications of atherosclerosis occur usually in middle and late age. Yet, the development of atherosclerosis may begin early in life, and the asymptomatic subclinical phase may last for decades. Early atherosclerotic changes, such as the thickening of vessel wall and alterations in endothelial functions, are the indicators of subclinical atherosclerosis that can be detected by various non-invasive imaging methods. The use of these methodologies has given information about the connections between subclinical atherosclerosis and the LDL-LOP.

Toikka et al. [53] investigated LDL oxidation among young healthy men with borderline hypertension. High-resolution ultrasound was used to measure the intima–media thickness (IMT) of the carotid and brachial arteries, cardiac dimensions, and brachial artery endothelial function. In this study, LDL-LOP was found to correlate positively with both the carotid and brachial IMTs. In multivariate analyses, the only predictors of the carotid IMT were the 24 h systolic blood pressure and the concentration of the LDL-LOP [53].

Another study from the same group investigated relationships between myocardial flow reserve, carotid IMT, and LDL-LOP in young men free from coronary heart disease [80]. The basal and stimulated coronary blood flow was measured using positron emission tomography (PET) in 55 healthy men, and the mean carotid artery IMT was measured using high-resolution ultrasound. In this study, the LDL-LOP concentration was found to correlate inversely with the flow reserve, and directly with the carotid IMT. These data suggested that an increased LDL-LOP level is directly related to early structural and functional atherosclerotic vascular changes.

In a study on the surrogate markers of asymptomatic atherosclerosis, Toikka et al. [49] investigated arterial elasticity, serum lipoproteins, and LDL-LOP in young healthy men and age-matched asymptomatic men with FH. As a marker of arterial elasticity, compliance in the thoracic aorta was measured by using magnetic resonance imaging, and in the common carotid artery by using ultrasound. It was found that the elasticity of both the carotid and brachial arteries was negatively associated with the concentration of the LDL-LOP [49]. In all, this was the first study to demonstrate an in vivo association between oxidized LDL and arterial elasticity.

### 9.7. Clinically Verified Atherosclerosis

The relation between LDL-LOP and coronary atherosclerosis was investigated in a sample of 62 men who underwent a diagnostic coronary angiography and sonography to measure the carotid IMT [52]. It was found that the LDL-LOP/LDL-cholesterol ratio and the carotid IMT were the only factors associated independently with the severity of coronary atherosclerosis. Moreover, the patients with a multi-vessel disease who did not use lipid-lowering therapy had 41% higher LDL-LOP/LDL-cholesterol ratio than the patients with normal vessels.

The finding of the association between the LDL-LOP and coronary atherosclerosis was confirmed and extended to the HDL-LOP in a study based on the measurement of the coronary plaque parameters by CT angiography [64]. While the LDL-LOP concentration had a significant positive and the HDL-LOP a negative association with the non-calcified plaque burden, the traditional lipid risk factors showed no associations.

### 9.8. Mortality

The predictive role of LDL-LOP in all-cause mortality was investigated in a population of 1260 elderly (aged 64 years or more) inhabitants [81]. They participated in the study in 1998–1999, and the medical records were re-examined one decade later. During the ten-year period 467 (37%) participants had died, and in 36% of the cases the cause of death was atherosclerotic and/or ischemic CVD. Comparisons between the survivors and deceased subjects showed that the total levels of the LDL-LOP did not differ between the groups. However, when proportioned to LDL-cholesterol, HDL-cholesterol or apo A-1, the LDL-LOP concentration stood out as a predictor of all-cause mortality. This finding was independent of age, sex, BMI, smoking, blood pressure, and diabetes.

### 9.9. Statin Treatment

Cholesterol lowering therapy is a major remedy in the treatment of the atherosclerotic vascular diseases, and the favorable effects of statins in primary and secondary prevention are well documented [82,83].

The effects of statins on LDL-LOP have been investigated in three different studies (Figure 5). The first study was a comparison between the effects of a Mediterranean-type diet and simvastatin treatment [71]. The trial was conducted in previously untreated hypercholesterolemic men aged 35 to 64 years. Simvastatin was given for 12 weeks at a dose of 20 mg/day. This treatment decreased the total and LDL-cholesterol concentrations by 21% and 30%, respectively. The concentration of the LDL-LOP was also decreased (by 16%). However, when the LDL-LOP value was proportioned to LDL-cholesterol, it appeared that the LDL-LOP/LDL-cholesterol ratio in fact was increased by 13% [71]. This means that despite the fact that the total amount of LDL-LOP in serum decreased due to the simvastatin treatment, the concentrations of the LDL-LOP in the remaining LDL particles increased.

Another study investigated the effects of 12-month simvastatin and atorvastatin therapies on LDL-LOP in patients with hypercholesterolemia and CVD [84]. The initial daily dose for both statins was 20 mg. The dose was doubled at 12 weeks if the subjects had not achieved the predefined lipid goals (LDL-cholesterol < 2.4 mmol/L; serum triglycerides < 1.5 mmol/L). After 52 weeks’ treatment LDL-cholesterol had decreased 47% by simvastatin and 50% by atorvastatin. Transient reductions in serum LDL-LOP concentrations were observed by both statins during the first 12 weeks’ treatment, but after the 52 week treatment period LDL-LOP did not differ from the baseline level in either statin group. This study confirmed the results of the previous study by Jula et al. [71] regarding the increase in the LDL-LOP/LDL-cholesterol ratio. After 52 weeks the ratio was substantially increased by both simvastatin (52%) and atorvastatin (59%) treatments [84].

The effects of pravastatin have been investigated in a study with 42 young normocholesterolemic subjects with type-1 diabetes [85]. Pravastatin was given to study subjects at a dose of 40 mg/day. The glycaemic control and insulin sensitivity remained unchanged during the 4 month treatment period. Pravastatin decreased the LDL-LOP concentration by 18%, and similarly to the treatments with the other statins pravastatin increased the ratio of LDL-LOP/LDL-cholesterol (by 32%).

## 10. LOP and Cholesterol: Commonalities and Disparities

Studies on LOP and the risk of atherosclerosis, as described in the previous chapter, indicate that the attributes of the lipoprotein LOP as a potential risk factor largely parallel those of cholesterol. As in the case of cholesterol, high LDL-LOP is associated with an increased risk and high HDL-LOP with protection. Lipoprotein LOP is related to blood pressure and BMI like cholesterol, and the physically active lifestyle and caloric restriction have similar effects on both the lipoprotein LOP and cholesterol levels. Due to the common transport and cellular intake mechanisms, however, the available data do not allow the elucidation of the independent effects of either cholesterol or LOP.

The age dependency of the lipoprotein-contained LOP and cholesterol, in turn, was found to be different. In the case of the LOP, the concentration is reported to increase in LDL and decrease in HDL with age. While there appears to be no clear trend for a change in the LDL-cholesterol, the concentration of the HDL-cholesterol increases with age [51]. The elevated LDL-LOP/HDL-LOP ratio may mean increased pro-oxidant burden and a risk of atherosclerotic changes with increasing age.

The most striking differences between the lipoprotein LOP and cholesterol are seen in their acute responses to high-fat meals and physical exercise. Both of these have notable influences on the levels of the lipoprotein LOP: a fatty meal increases for several hours the concentration of LDL-LOP, and strenuous physical exercise elevates substantially the concentration of HDL-LOP. The effects of high-fat meals and exercise on lipoprotein cholesterol levels are negligible.

The different responses of LDL-LOP and LDL-cholesterol to a fatty meal demonstrate that, unlike cholesterol, LOP in food may rapidly affect the amount of the atherogenic substances in LDL. It should be kept in mind that LOP comprises molecules which have no physiological role and are capable of directly initiating and boosting atherogenic processes. The exercise-induced increase in HDL-LOP, on the other hand, may be indicative of a protective LOP-clearing function of HDL and also of the ability of the HDL to respond rapidly to increased LOP levels in the body.

Yet another disparity between the lipoprotein LOP and cholesterol is seen in their responses to statin treatment. Three independent studies on statins and LDL-LOP have uniformly shown that statin treatment is less effective in reducing the concentration of LDL-LOP than that of LDL-cholesterol, which results in the elevation of the LDL-LOP/LDL-cholesterol ratio. The apparent discrepancy may indicate that the LOP in LDL impairs the LDL receptor-mediated cellular intake of the LDL particles. Another possible explanation is that while statin accelerates the LDL receptor-mediated removal of the native LDL particles form the circulation, the rate of the scavenger receptor-mediated removal of the oxidized LDL is not changed [84]. Nevertheless, a higher amount of LOP in the remaining LDL particles could mean an increased atherogenicity of the circulating LDL. It is well known that CVD events may remain prevalent also during high-intensity statin therapy [86]. This so-called “residual risk” appears to depend on lipid and/or lipoprotein factors [87], and it is tempting to speculate that the enrichment of LOP in LDL, caused by statin treatment, could in part explain the residual risk.

## 11. Lipoprotein LOP as a Potential Risk Factor

The LOP in lipoproteins have usually been regarded as a consequence of the endogenous oxidative stress, but the presented evidence of the lipoprotein-specific LOP transport is suggesting a role of lipoprotein LOP as a risk factor of atherosclerosis. It is therefore worth considering how this would affect the conception and management of the disease.

### 11.1. LOP as Dietary Risk Factor

The oxidative modification of the LDL has long been assumed to result from endogenous free radical reactions [11]. It has come out, however, that a fatty meal increases the amount of LOP in LDL, which is capable of causing atherogenic LDL modifications. These findings draw attention to another type of dietary risk factors, LOP formed, e.g., during food storage, processing, or digestion, emphasizing the importance of the pro/antioxidant effects of nutrients. In support of this view, experimental evidence shows that the postprandial LDL is more atherogenic than the LDL in the post absorptive state. The more pronounced atherogenicity was postulated to be due to lipid peroxides in the postprandial LDL [88].

A hamburger meal, the LOP content of which was reported to be close to 1 mmol, was sufficient to double the concentration of the LOP in postprandial LDL [9]. In UK the average daily intake of lipid hydroperoxides associated with fats and oils was estimated to be approximately 1.5 mmol per day [31]. It is therefore possible that at the average level of fat consumption a substantial part of the LDL-LOP could be of dietary origin. This applies particularly to the postprandial situation, and except for the early morning, most individuals are in the nonfasting state most of the time. If the conclusion of the dietary origin of the LOP is tenable, then LOP in food should be considered as a notable risk factor for CVD. It would also mean that LDL is an important mediator of the risk.

### 11.2. HDL and the Clearance of LOP

According to the conception of the lipoprotein-specific LOP transport, HDL-assisted reverse transport carries LOP from peripheral veins and tissues towards the liver. This conclusion is based on studies showing that physical exercise affects the LDL-LOP/HDL-LOP balance in such a way that the net flow of LOP shifts towards the liver. Together with the previous observations of the capability of the liver to remove and eliminate LOP in HDL, and results from studies with apoA1-mimetic peptides, these findings suggest a protective LOP-clearing function for HDL. This is an interesting new perspective for the discussion of the functionality of HDL. HDL-LOP is also reported to be associated with a wide spectrum of atherosclerosis risk factors. In most cases these associations are opposite to those of the LDL-LOP, giving further support for the protective role of HDL.

## 12. Significance for Other Oxidative Stress-Dependent Diseases

As the carriers of the LOP in circulation, lipoproteins likely contribute to the pro/antioxidant balance in the body. This raises the question whether the lipoprotein-specific transport of LOP would be implicated in oxidative stress-dependent pathological processes other than atherosclerosis. Yet, it must be kept in mind that there are no common criteria for the “oxidative stress-dependent” diseases and/or pathological processes, and it may sometimes be difficult to differentiate whether oxidative stress is a cause or consequence of the disease process.

In a previous chapter (Metabolic disorders related to atherosclerosis) data were presented about the associations of the lipoprotein LOP with the development of fatty liver, which also associates with oxidative stress [89]. Similarly, data were presented regarding a connection between lipoprotein LOP and insulin sensitivity. Lipoprotein LOP was found to be involved in both these pathophysiological situations. Besides these reported studies, there appear to be no investigations on lipoprotein LOP in diseases or pathological processes in which oxidative stress is implicated.

Type 2 diabetes, which is a major worldwide public health problem today, is an example of the diseases in which the contribution of the lipoprotein LOP transport would be worth investigating. The involvement of oxidative stress in type 2 diabetes has been demonstrated by the biomarkers of the oxidative lipid, protein, and nucleic acid damage [90]. Another disease of emerging worldwide concern, chronic kidney disease (CKD), could also be of interest in this respect. Oxidative stress and inflammation are important contributors in the pathogenesis and progression of CKD [91,92,93]. In both these diseases dyslipidemia, too, appears to play an important role [94,95,96]. It is possible that together with the oxidative stress, unfavorable lipoprotein levels could enhance the development of these pathophysiological conditions.

## 13. Perspective Regarding Prevention and Treatment

If the lipoprotein-specific transport of the LOP plays a role in the development of atherosclerotic CVD, then the strategies for prevention and treatment should aim at low LDL-LOP/HDL-LOP ratio by lowering LDL-LOP and/or elevating HDL-LOP levels. As discussed in a previous chapter, caloric restriction, physically active lifestyle, and statin treatment have been shown to improve LDL-LOP effectively. Therefore, since these current strategies for a healthier lipid profile also decrease the potential risk due to LOP, the role of LOP as a risk factor would hardly radically change the management of CVD.

On the other hand, as it has turned out that the amount and composition of the circulating LOP, especially during the postprandial period, is largely determined by ingested food, LOP in food could be regarded as a new type of dietary risk factor. Shifting the focus from cholesterol towards LOP could therefore give additional tools for prevention and therapy. This could mean in the future, e.g., the reporting of the presence of LOP in foodstuffs, similarly to the present practice for the content and quality of fat.

In line with the postulated role of the LOP as a risk factor, epidemiological studies show that low levels of the dietary antioxidants are associated with an increased risk of CVD and that increased intakes are protective [97]. Yet, antioxidant supplementation is not an answer since, as discussed earlier (in Dietary factors), it does not affect the concentration of LDL-LOP even though it substantially increases the antioxidant capacity of the LDL. This is well in agreement with the poor success of the antioxidant strategies in limiting atherosclerosis and CVD [11], and supports the idea of the dietary origin of the LOP in LDL. Dietary supplements other than antioxidants could, however, be useful as reported in the studies with separate components of nutrition (See Section 9.4).

In recent years, the development of new strategies for the prevention and treatment of CVD has increasingly focused on HDL and its functions [14,15]. Despite the huge investments on research and drug development projects, attempts to reinforce and exploit the protective effects of the HDL as clinical applications have turned out to be challenging. There is little information concerning possibilities to enhance the LOP transporting capacity of the HDL. It has been reported that a 6-month supervised physical activity program increased the concentration of HDL-LOP and the HDL-LOP/HDL-cholesterol ratio among elderly women [59]. It is obvious that a deeper understanding of factors specifically determining the transport capacity is needed before it is possible to modulate this function.

Since the studies of the LOP transport function are based on the measurement of the LOP in lipoproteins, it would be worth considering whether assays for the lipoprotein LOP could form a basis for new diagnostic applications. Circulating LDL-LOP have been measured over the past 25 years in a variety of cross-sectional, intervention, and population-based studies, and it appeared early that the measure of the LDL-LOP is an applicable indicator of the risk of CVD [49]. In these studies, high LDL-LOP has consistently pointed to an increased risk of the disease (See Lipoprotein-specific transport of LOP and the risk of atherosclerosis). The assay for HDL-LOP was introduced much later, but studies thus far have shown that a low HDL-LOP level is associated with an elevated risk. The ratio of LDL-LOP/HDL-LOP, as a putative indicator of the direction of the net movement of the circulating LOP, has proved to be potentially interesting and worth further investigation.

Even though the available evidence supports the usability of the lipoprotein LOP as an indicator of the risk of atherosclerosis and CVD, large-scale epidemiological studies are needed for verification. In addition, the present methodologies need to be developed to meet the requirements of modern clinical laboratories. Malondialdehyde (MDA) and other reactive LOP are known to form adducts on proteins or lipids of lipoproteins [11,98], which can serve as the indicators of the oxidative stress. Yet, due to the firm attachment on the structural components of the lipoproteins they are not indicative of the LOP transport function. Similarly, the common assay for oxidized LDL, which is a measure of the oxidation-dependent aldehyde substitution of the lysine residues in apolipoprotein B, is indicative of the oxidative stress but not of the LOP transport function.

## 14. Consistence with Present Theories

How does the lipoprotein-specific transport of the LOP comport with the present view of the etiology of the atherosclerotic CVD? Various theories have been developed to explain the etiology of the disease, each of them emphasizing a specific part in the processes of atherogenesis [11]. Since the discoveries by Goldstein and Brown [99], the role of the lipoproteins as risk factors for atherosclerosis has been explained by the cholesterol transport function. More recently, the oxidation theory of atherosclerosis, and the oxidative response to inflammation hypothesis [11] stressing the role of monocytes and vascular inflammation, have further shaped the view.

The proposed role of the LOP transport function does not conflict with any of these theories, and rather underlines the significance of both the lipoprotein transport and oxidation. Moreover, since LOPs are implicated in the accumulation of monocytes in the subendothelial space, the hypothesis of the role of the lipoprotein LOP transport is compatible also with the oxidative response to inflammation hypothesis. Taken together, the proposed role of the lipoprotein-specific LOP transport principally agrees with the previously presented hypotheses, and, in fact, not only expands but also combines the previous theories in a rational way. The only deviation concerns the origin of the “oxidized LDL”. Thus far it has been assumed that “oxidized LDL” is generated within the vessel wall in reactions caused by endogenous free radicals [11]. As an addition to this, the theory emphasizing the role of the lipoprotein LOP transport proposes that part of the lipoprotein LOP would be of dietary origin. This argument is reasoned by evidence from the dietary studies with fatty meals.

## 15. Open Questions and Future Directions

The fact that lifestyle-dependent factors affect in a parallel way both the cholesterol and LOP levels suggests that the lipoprotein-specific LOP transport at least in part works under the same regulatory control as the cholesterol transport. Unlike in the case of cholesterol, there is at present no information regarding the influence of the LOP on the regulation of the lipoprotein metabolism.

The molecular mechanisms of the transfer of cholesterol from cells and tissues to HDL, between lipoproteins, and from lipoproteins to the liver are today well known, but is not known whether the same mechanisms apply also for LOP. To increase the credibility of the lipoprotein-specific LOP transport hypothesis, it would be important to know the molecular mechanisms of the LOP transfer at various steps of the transportation. In the current situation it is not known whether drug development for the lowering of the lipoprotein LOP levels could follow the same paths as in the case of cholesterol. Increasing the knowledge would help to find out whether there are ways to improve the situation beyond the strategies used in the cholesterol-targeted treatment protocols.

During the last three decades the topic of this review, lipoprotein LOP transport, has received surprisingly little attention in the study of atherosclerosis, even though the number of studies on “oxidized LDL” has at the same time vastly increased. As discussed in a previous chapter (Perspective Regarding Prevention and Treatment), it should be recognized that the measurement of the “oxidized LDL” by conventional methods (other than LOP) is not indicative of the lipoprotein transport function. Due to the limited number of active research groups in this specific field, a substantial part of the studies cited as the evidence of the lipoprotein LOP transport come from one research center, and this may be seen as a weakness of the study. Therefore, it would be most important to broaden the network of investigators and have more evidence based on research. Moreover, expanding studies to diverse populations is essential to confirm whether the results are applicable to the general population. An advancement that could help to boost the development would be the availability of the methodology that is suitable for large-scale epidemiological studies and applicable in modern clinical laboratories.

The multiplicity of the LOPs, which are an assortment of chemicals with various structures, forms a challenge for attempts to control the possible risks. The diverse constituents of the LOP presumably differ from each other regarding their biological effects, and the research should therefore be directed to the characterization of the role of individual chemicals and chemical groups within the LOP.

Another place where chemical analytics will be needed is the monitoring of the LOP in foodstuffs. A reduction in the dietary intake of the LOP will not be possible without the knowledge of the presence of the LOP in dietary components. Thus far, the data concerning LOP in food have come out randomly with no aspiration to develop a systematical survey.

## 16. Conclusions

The evidence connecting the lipoprotein LOP transport to atherosclerosis is versatile and consistent, revealing surprisingly strong connections between the transport of LOP and the risk of atherosclerotic CVD. If confirmed, the proposed LOP transport function would open a new and interesting viewpoint for the study of atherosclerosis. Importantly, the proposed role of the lipoprotein-specific LOP transport is in full agreement with the existing theories of the development of atherosclerosis.

The data presented in this review show that the attributes of the lipoprotein LOP as a potential risk factor parallel those of cholesterol, and due to the common transport and cellular intake mechanisms the elucidation of their independent effects is difficult. Keeping in mind that LOP, unlike cholesterol, can directly initiate and boost atherogenic processes, it appears likely that the knowledge of the lipoprotein-specific LOP transport will affect the present view of the role of cholesterol as a risk factor.

If confirmed, the proposed lipoprotein LOP transport function would open new perspectives regarding the risk of CVD, such as the dietary origin of LOP, and the protective function of HDL in the clearance of LOP. Focusing on LOP could give additional tools especially for prevention and diagnosis but would hardly radically change the management of CVD.

## Figures and Tables

**Figure 1 antioxidants-13-00512-f001:**
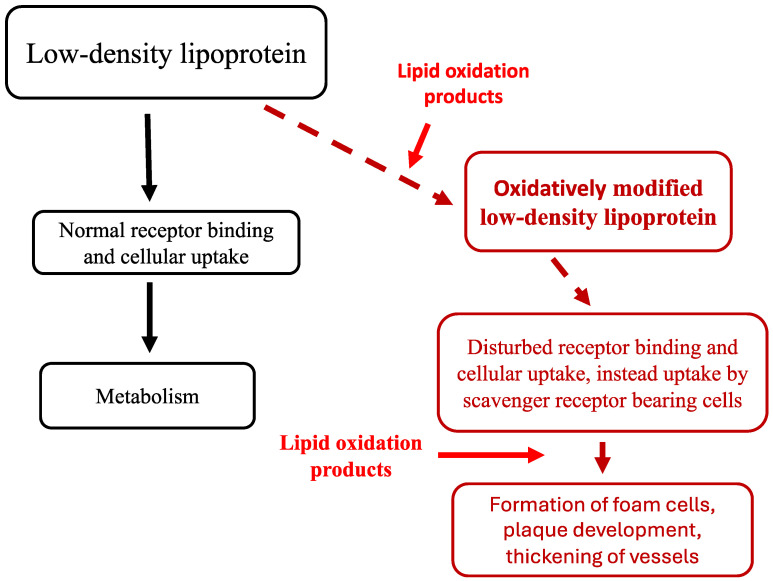
Proposed roles of LOP in the initiation and progression of atherosclerosis. Oxidative modification on LOP disturbs the normal receptor binding and metabolism of LDL. LOP can also by several mechanisms stimulate the subsequent development of foam cells, plaque development, and the thickening of vessel walls.

**Figure 2 antioxidants-13-00512-f002:**
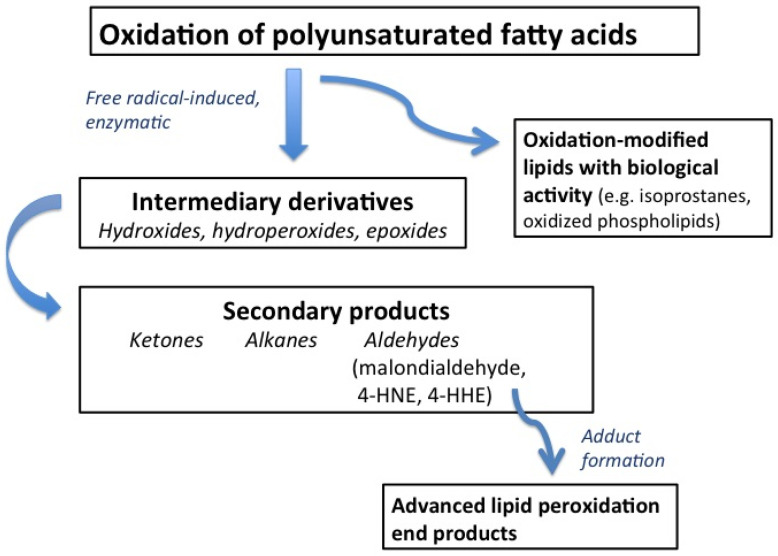
Formation of LOP during peroxidation of polyunsaturated fatty acids.

**Figure 3 antioxidants-13-00512-f003:**
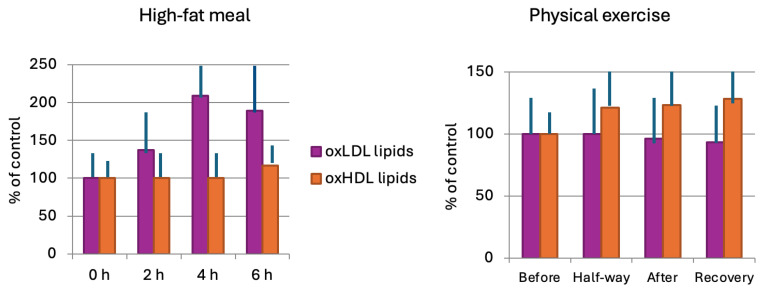
Effects of high-fat meal and strenuous physical exercise on lipoprotein LOP. High-fat meal (**left panel**): blood samples were taken before, and 2, 4, and 6 h after the consumption of the meal (a standard hamburger meal) and 4 dL of fruit juice. Physical exercise (**right panel**): The subjects performed on a treadmill a 40 min tempo run at the velocity corresponding to 80% VO_2max_. Blood samples were taken before, in the middle, and immediately after the exercise, as well as after a 90 min recovery period. Based on data from [9].

**Figure 4 antioxidants-13-00512-f004:**
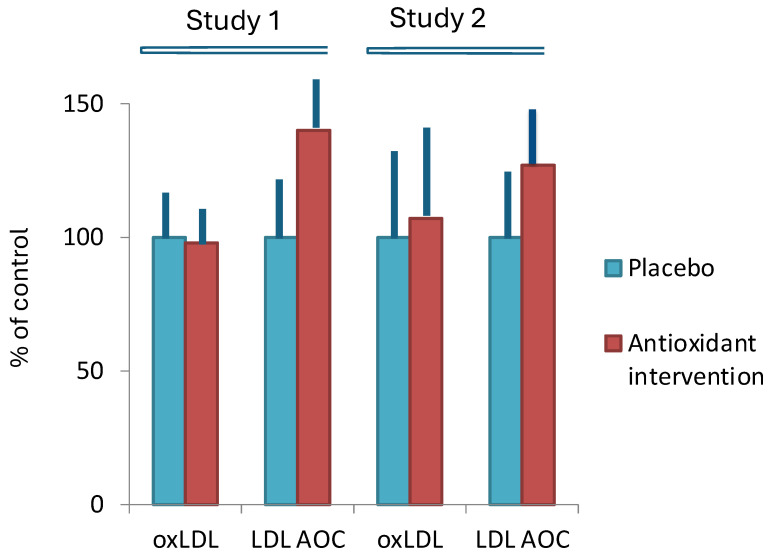
Effects of antioxidant intervention on LDL-LOP and LDL antioxidant capacity (LDL-AOC) in healthy volunteers. Study 1: daily doses of 294 mg 100 mg d-α-tocopherol acetate, 1000 mg of ascorbic acid, and 60 mg ubiquinone for 4 weeks. Study 2: daily doses of 100 mg d-α-tocopherol acetate and 500 mg of ascorbic acid for 8 weeks. Based on data from [61] (Study 1) and [74] (Study 2).

**Figure 5 antioxidants-13-00512-f005:**
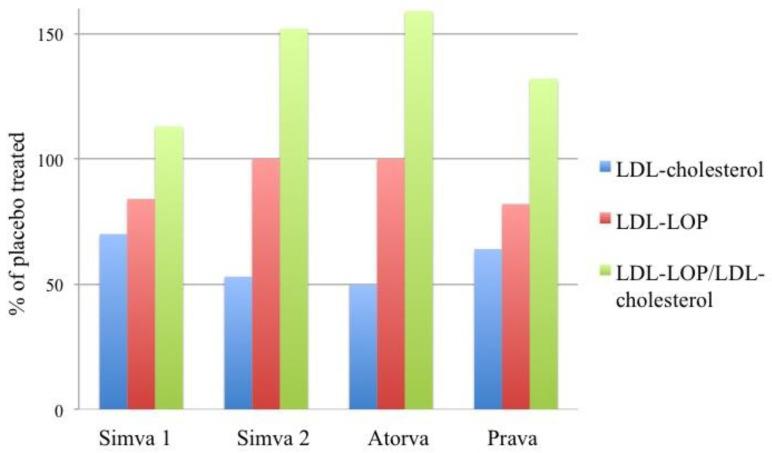
Effect of statin treatment on LDL-cholesterol, LDL-LOP, and the ratio of LDL-LOP/LDL-cholesterol. Statin treatments were investigated with three different statins in three separate studies. In the first study [71], simvastatin was given for 12 weeks at a dose of 20 mg/day (“Simva 1”). In the second study [84], simvastatin (“Simva 2”) and atorvastatin (“Atorva”) were given for 12 months at a dose that was initially 20 mg/day. The dose was doubled at 12 weeks if the subjects had not achieved predefined lipid goals. Subjects in the third study [85] were given pravastatin (“Prava”) for 4 months at a dose of 40 mg/day.

**Table 1 antioxidants-13-00512-t001:** Adverse effects of LOP.

macrophage foam cell formation
chemotactic for monocytes, chemostatic for tissue macrophages
cytotoxic, induction of apoptosis
mitogenic for smooth muscle cells and macrophages
platelet aggregation and thrombosis
pathologic angiogenesis

(Adapted from information provided in [12,14]).

**Table 2 antioxidants-13-00512-t002:** HDL functions.

Transport ○ cholesterol ○ fat-soluble vitamins ○ hormones ○ lipid oxidation products ○ xenobiotics ○ micro-RNAs
Cell communication ○ cholesterol efflux ○ signal transduction ○ endocytosis
The elimination of biohazards ○ bacterial toxins ○ lipid oxidation products

(Adapted from information provided in [15]).

**Table 3 antioxidants-13-00512-t003:** Findings of CVD risk factors and lipoprotein LOP.

Influence of/Association with	Oxidized LDL Lipids	Oxidized HDL Lipids
Age	Increases with age	Decreases with age
Gender	Male > female	Female > male
Body mass index	Positive association	Negative association
Weight reduction	Substantial decrease	Not studied
Tobacco smoking	Elevated among smokers	Not studied
Statin treatment	Substantial decrease	
Acute physical exercise	No change/decrease	Increase
Physically active lifestyle	Substantial decrease	Slight increase
Fatty meal	Substantial increase within 2 h	Slight increase after several hours
Saturated fat	Positive association	Not studied
Polyunsaturated fat	Negative association with n-6 PUFA	Not studied
Atherosclerosis	Positive association with brachial, carotid, and coronary atherosclerosis	Negative association with coronary atherosclerosis
Insulin resistance	Positive association	Not studied
Metabolic syndrome	Positive association	Not studied
Development of fatty liver	Positive association	Negative association

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
