# Peer review of "Lipid Oxidation Products and the Risk of Cardiovascular Diseases: Role of Lipoprotein Transport"

_antioxidants, 2024, doi:10.3390/antiox13050512_

Round 1

Reviewer 1 Report

The author summarizes research, mainly by his group in Finland, on lipid oxidation products (LOP) transported by lipoproteins in blood and their relation to atherosclerosis progression.  This research has yielded some important information and insights, most notably that LOP may be a major--or even primary--factor contributing to atherogenesis, that lipid oxidation during food preparation (especially cooking/frying), rather than endogenous oxidative processes, is likely the main cause of LOP transported in blood, that the importance of HDL mainly resides in its transport of LOP to the liver, and that LOP may be largely responsible for residual CVD risks despite aggressive LDL cholesterol-lowering therapy.  Although the research results summarized are almost certainly applicable to the general population, a limitation of these studies is that they were done exclusively on Finnish populations, which are ethnographically distinct and even unique.  This is unavoidable, since Dr. Ahotupa's group of investigators have developed the methods for measuring LOP, which apparently are not being employed elsewhere.  Although the suggestion that similar studies should be undertaken by other investigators is advanced in the Discussion, perhaps this point could be made more pointedly.  Nevertheless, this review is a valuable contribution to the field that should be published.

Awkward English usage (not spelling or formatting), e.g., line 115, including missing articles.  Y-axis labels (units), as well as variance data, missing from Figures 2 and 3.

Author Response

Responses to Reviewer 1

The author summarizes research, mainly by his group in Finland, on lipid oxidation products (LOP) transported by lipoproteins in blood and their relation to atherosclerosis progression.  This research has yielded some important information and insights, most notably that LOP may be a major--or even primary--factor contributing to atherogenesis, that lipid oxidation during food preparation (especially cooking/frying), rather than endogenous oxidative processes, is likely the main cause of LOP transported in blood, that the importance of HDL mainly resides in its transport of LOP to the liver, and that LOP may be largely responsible for residual CVD risks despite aggressive LDL cholesterol-lowering therapy.  Although the research results summarized are almost certainly applicable to the general population, a limitation of these studies is that they were done exclusively on Finnish populations, which are ethnographically distinct and even unique.  This is unavoidable, since Dr. Ahotupa's group of investigators have developed the methods for measuring LOP, which apparently are not being employed elsewhere.  Although the suggestion that similar studies should be undertaken by other investigators is advanced in the Discussion, perhaps this point could be made more pointedly.  Nevertheless, this review is a valuable contribution to the field that should be published.

Detail comments

Awkward English usage (not spelling or formatting), e.g., line 115, including missing articles.  Y-axis labels (units), as well as variance data, missing from Figures 2 and 3.

Response of the author:

Thank you for the comments and the constructive criticism. The manuscript has been revised by an experienced English-speaking colleague, and the Figures were corrected as suggested. A comment concerning applicability of the results to the general population has been added to Chapter 15 ("Open questions and future directions", lines 1010-1011).

Reviewer 2 Report

This review discusses the utility of LOP in cardiovascular diseases. While the review enhances understanding of LOP, several areas for improvement have been identified.

1. The relationship between LOP and oxidized LDL is complex and challenging to comprehend. Therefore, it is necessary for the author to clearly summarize the differences and relationship between LOP and oxidized LDL at the beginning of the review.

2. To facilitate readers' understanding of LOP, the author needs to provide a specific explanation of what LOP is.

3. When explaining the relationship between LOP and oxidized LDL, the use of figures should be employed to facilitate visual understanding.

4. In this review, there is also mention of the potential impact of LOP on cardiovascular diseases in the future. Just as the decrease in oxidized LDL has been reported through supplements (Funamoto M, et al. Int J Chron Obstruct Pulmon Dis. 2016 Aug 26;11:2029-34), etc., it should be mentioned in comparison with oxidized LDL whether there is a possibility for LOP to change through supplements, diet, etc.

This review discusses the utility of LOP in cardiovascular diseases. While the review enhances understanding of LOP, several areas for improvement have been identified.

1. The relationship between LOP and oxidized LDL is complex and challenging to comprehend. Therefore, it is necessary for the author to clearly summarize the differences and relationship between LOP and oxidized LDL at the beginning of the review.

2. To facilitate readers' understanding of LOP, the author needs to provide a specific explanation of what LOP is.

3. When explaining the relationship between LOP and oxidized LDL, the use of figures should be employed to facilitate visual understanding.

4. In this review, there is also mention of the potential impact of LOP on cardiovascular diseases in the future. Just as the decrease in oxidized LDL has been reported through supplements (Funamoto M, et al. Int J Chron Obstruct Pulmon Dis. 2016 Aug 26;11:2029-34), etc., it should be mentioned in comparison with oxidized LDL whether there is a possibility for LOP to change through supplements, diet, etc.

Author Response

Responses to Reviewer 2

Major comments 

This review discusses the utility of LOP in cardiovascular diseases. While the review enhances understanding of LOP, several areas for improvement have been identified. 

1. The relationship between LOP and oxidized LDL is complex and challenging to comprehend. Therefore, it is necessary for the author to clearly summarize the differences and relationship between LOP and oxidized LDL at the beginning of the review. 

Response of the author:

This is indeed an important notice. For clarification of the relationship a new Figure 1 (page 3), and text describing the relationship (Introduction lines 80-84), have been added.

2. To facilitate readers' understanding of LOP, the author needs to provide a specific explanation of what LOP is. 

Response of the author:

This is now explained in the revised version (Introduction lines 73-74).

3. When explaining the relationship between LOP and oxidized LDL, the use of figures should be employed to facilitate visual understanding. 

Response of the author:

See the response to the first comment.

4. In this review, there is also mention of the potential impact of LOP on cardiovascular diseases in the future. Just as the decrease in oxidized LDL has been reported through supplements (Funamoto M, et al. Int J Chron Obstruct Pulmon Dis. 2016 Aug 26;11:2029-34), etc., it should be mentioned in comparison with oxidized LDL whether there is a possibility for LOP to change through supplements, diet, etc. 

Response of the author:

As a response to this comment one sentence was added to Chapter 13 ("Perspective regarding prevention and treatment", lines 926-928).

Detail comments 

This review discusses the utility of LOP in cardiovascular diseases. While the review enhances understanding of LOP, several areas for improvement have been identified. 

1. The relationship between LOP and oxidized LDL is complex and challenging to comprehend. Therefore, it is necessary for the author to clearly summarize the differences and relationship between LOP and oxidized LDL at the beginning of the review. 

2. To facilitate readers' understanding of LOP, the author needs to provide a specific explanation of what LOP is. 

3. When explaining the relationship between LOP and oxidized LDL, the use of figures should be employed to facilitate visual understanding. 

4. In this review, there is also mention of the potential impact of LOP on cardiovascular diseases in the future. Just as the decrease in oxidized LDL has been reported through supplements (Funamoto M, et al. Int J Chron Obstruct Pulmon Dis. 2016 Aug 26;11:2029-34), etc., it should be mentioned in comparison with oxidized LDL whether there is a possibility for LOP to change through supplements, diet, etc.

Response of the author:

Responses to the Detail comments are given with the Major comments above.

Round 2

Reviewer 2 Report

OK

OK